# Harnessing Porphyrin Accumulation in Liver Cancer: Combining Genomic Data and Drug Targeting

**DOI:** 10.3390/biom14080959

**Published:** 2024-08-07

**Authors:** Swamy R. Adapa, Pravin Meshram, Abdus Sami, Rays H. Y. Jiang

**Affiliations:** 1USF Genomics Program, Center for Global Health and Infectious Diseases, College of Public Health, University of South Florida, Tampa, FL 33612, USA; swamyrakesh@usf.edu; 2Global and Planetary Health, College of Public Health, University of South Florida, Tampa, FL 33612, USA; pravinsdeesha@gmail.com; 3Department of Molecular Medicine, Morsani College of Medicine, University of South Florida, Tampa, FL 33612, USA; abdussami@usf.edu

**Keywords:** heme, porphyrin, protoporphyrin IX, liver, P450, metabolism, liver cancer

## Abstract

The liver, a pivotal organ in human metabolism, serves as a primary site for heme biosynthesis, alongside bone marrow. Maintaining precise control over heme production is paramount in healthy livers to meet high metabolic demands while averting potential toxicity from intermediate metabolites, notably protoporphyrin IX. Intriguingly, our recent research uncovers a disrupted heme biosynthesis process termed ‘porphyrin overdrive’ in cancers that fosters the accumulation of heme intermediates, potentially bolstering tumor survival. Here, we investigate heme and porphyrin metabolism in both healthy and oncogenic human livers, utilizing primary human liver transcriptomics and single-cell RNA sequencing (scRNAseq). Our investigations unveil robust gene expression patterns in heme biosynthesis in healthy livers, supporting electron transport chain (ETC) and cytochrome P450 function without intermediate accumulation. Conversely, liver cancers exhibit rewired heme biosynthesis and a massive downregulation of cytochrome P450 gene expression. Notably, despite diminished drug metabolism, gene expression analysis shows that heme supply to the ETC remains largely unaltered or even elevated with patient cancer progression, suggesting a metabolic priority shift. Liver cancers selectively accumulate intermediates, which are absent in normal tissues, implicating their role in disease advancement as inferred by expression analysis. Furthermore, our findings in genomics establish a link between the aberrant gene expression of porphyrin metabolism and inferior overall survival in aggressive cancers, indicating potential targets for clinical therapy development. We provide in vitro proof-of-concept data on targeting porphyrin overdrive with a drug synergy strategy.

## 1. Introduction

The liver, a central hub of metabolic activity in the human body, exerts important influence over vital physiological functions, such as detoxifying harmful substances and producing bile for digestion [1]. Among its multifaceted duties, the liver assumes a pivotal role (alongside bone marrow) in heme production, a cornerstone process crucial for a myriad of cellular functions, including energy generation and detoxification. This interplay between heme and liver function is evident in various liver diseases linked to heme metabolism [1]. One such condition is porphyria, which comprises a spectrum of disorders stemming from aberrations in heme production [2,3]. Liver cirrhosis, characterized by progressive liver scarring and dysfunction, also perturbs heme metabolism due to compromised liver integrity [4]. Additionally, inherited metabolic disorders such as Wilson’s disease and hemochromatosis can disrupt heme homeostasis, precipitating liver-related complications [5,6]. These disease manifestations demonstrate the inseparable relationship between heme metabolism and liver health.

In normal hepatic physiology, heme biosynthesis stands out as a meticulously regulated process, characterized by the robust production of cytochrome P450 (P450) enzymes [7,8]. Cytochrome P450 enzymes, often abbreviated as CYPs, are a superfamily of heme-containing enzymes found primarily in the liver [9,10,11,12,13,14,15]. The name “cytochrome P450” comes from the fact that these enzymes absorb light at a wavelength of 450 nm when bound to carbon monoxide, an optical property unique to heme molecules in CYPs [16,17]. These enzymes play a central role in various hepatic pathways, including drug metabolism, steroid synthesis, and the detoxification of endogenous and exogenous compounds [18]. Moreover, P450 enzymes are key players in biotransformation of numerous drugs, influencing their efficacy and toxicity profiles [7,8]. Genetic polymorphisms in cytochrome P450 genes can lead to interindividual variability in drug metabolism and response. Individuals carrying certain allelic variants may metabolize drugs at different rates, affecting drug efficacy and toxicity [19]. This P450-related phenomenon demonstrates the importance of personalized medicine approaches in drug therapy [15,20,21].

In the liver’s normal physiological operation, a significant reservoir of heme is essential to sustain the electron transport chain (ETC), a fundamental component driving cellular energy production. This intricate molecular machinery oversees the synthesis of adenosine triphosphate (ATP) through oxidative phosphorylation, which is crucial for powering cellular processes. However, conflicting observations have emerged regarding the status of the ETC in oncogenic processes. While certain studies propose dysfunctions within the mitochondrial apparatus and potential defects in the ETC among cancerous tissues [22,23], recent investigations have revealed a paradoxical phenomenon: an upsurge in ETC activity within the in vivo setting of cancers [24,25,26]. This duality in reported outcomes highlights the complexities of metabolic reprogramming in cancer [27] and emphasizes the need for further investigation into the interconnectedness of heme metabolism, ETC function, and oncogenesis.

In human oncogenesis, we have recently identified a heme-related metabolic rewiring phenomenon termed “porphyrin overdrive” [28], signifying a distinct metabolic shift in cancerous tissues. Our research demonstrates cancer’s genetic reliance on an upregulation of mid-pathway heme biosynthesis genes and the accumulation of intermediate porphyrins, a characteristic absent in healthy tissues. The healthy liver, as a primary site for mass heme synthesis, maintains precise biosynthetic control to ensure production while averting intermediate toxicity [1,4,29,30,31]. However, our finding suggests that this homeostasis is disrupted in cancer [28,31,32,33], leading to porphyrin accumulation crucial for oncogenesis. Recent studies also indicate that aberrant heme biosynthesis and export can regulate the TCA cycle and oxidative phosphorylation (OXPHOS), thereby enhancing oncogenic proliferation [34,35]. Nonetheless, the intricate relationship between liver heme metabolism and porphyrin production within the context of cancer remains largely enigmatic. We hypothesize that oncogenic livers, despite downregulating overall heme production, produce intermediate porphyrins, which may play an as-yet-undetermined role in supporting oncogenesis. Further exploration of this connection between porphyrin and heme may unveil novel therapeutic avenues.

## 2. Materials and Methods

### 2.1. Primary Liver Cell and Hepatoma Cell Line Preparation

Primary human hepatocyte cells, purchased from BioIVT, Westbury, NY, USA (Cat No. M00995-P), were utilized without any culturing steps. Two de-identified healthy donors, UBV (male) and PDC (male), provided primary human hepatocytes for the RNAseq experiments. Cryopreserved primary human hepatocytes were thawed by immersion in a 37 °C water bath for 2 min, followed by sterilization with 70% ethanol in a sterile field.

In addition, hepatoma cell lines HepG2 (ATCC #HB-8065), SNU449 (ATCC #CRL-2234), HUH7 (RRID:CVCL_0336), and HC-04 (kindly provided by Adam’s lab, University of South Florida, FL, USA) were utilized as liver cancer cell lines. These cryopreserved cell lines were thawed, suspended in the previously prepared hepatocyte culture medium, and transferred to T75 flasks coated with collagen (Thermal Fisher Scientific, Corning, MA, USA, Cat No. 354236) at a density of 5 μg/cm^2^. The HepG2 cell line medium was made by adding 10% fetal bovine serum (FBS, Sigma, St. Louis, MO, USA) and penicillin–streptomycin–neomycin (PSN) solution (100×, Cat No. 15640-055; Thermo Fisher Scientific, MA, USA) to Eagle’s Minimum Essential Medium (EMEM, Cat No. 30-2003, ATCC, Manassas, VA, USA). The SNU-449 and HUH-7 cell line medium was made by adding 10% FBS (Sigma-Aldrich) and penicillin–streptomycin–neomycin (PSN) solution (100×, Cat No. 15640-055; Thermo Fisher Scientific) to RPMI 1640 medium (Gibco, Waltham, MA USA). The HC-04 cell line culture medium was prepared by mixing the F12 base medium (Cat No. 11765-054; Invitrogen, Carlsbad, CA, USA,) and the MEM base medium (Cat No. A10490-01; Invitrogen) in a 1:1 ratio (*vol*/*vol*). This solution was supplemented with 10% fetal bovine serum (FBS; Cat No. SH30070; Hyclone, Logan, UT, USA), 1.0 M Hepes (Cat No. 15630-080; Invitrogen, Carlsbad, CA, USA,) and 200 mM glutamine (Cat No. 25030-081; Invitrogen, Carlsbad, CA, USA,). All the cell lines were grown until 70% confluence was reached, and the media were changed every other day. The cells were trypsin-hydrolyzed using TrypLE Express Enzyme (1×) (Cat. No. 12605028; Gibco) and washed using hepatocyte culture media. Cell counts were performed over time using the trypan blue exclusion method (Cat No. 25-900-CI; Corning). Total cell numbers were calculated based on the dilution factor at each time point using a disposable hemocytometer (Cat No. DHC-N01; INCYTO, Chungnam-do, 31056 Republic of Korea.

### 2.2. Bait-and-Kill Drug Assay

Hepatoma cells were cultured until 70% confluence was reached, and the culture media were changed every other day. Upon reaching the desired confluence, cells were trypsinized using TrypLE™ Express Enzyme (1×) (Gibco, Cat No. 12605028), washed with hepatocyte culture medium, and seeded at a density of 6000 cells/well in 384-well plates (Greiner Bio-One, Kremsmünster, Austria, Cat No. 781091), with 20 μL of medium per well. Cells were incubated at 37 °C for 4 h in very low light conditions, either in the absence or presence of 1.0 mM 5-aminolevulinic acid (ALA). Following incubation, cells were allowed to proliferate for the next 48 h under continued low light conditions. ALA-treated and untreated cells were plated in triplicate wells on 384-well plates for both the 4 h and 24 h treatments, with drugs in an 18-point concentration series, ranging from 132 µM to 1 nM, and a final volume of 25.6 μL per well.

Cell viability was assessed using the CellTiter-Glo^®^ 2.0 reagent (Promega, Madison, WI, USA, Cat No. G9243) to measure cellular ATP levels according to the manufacturer’s instructions, with 25.6 μL of reagent added per well. Luminescence was quantified using a Clariostar Plus Microplate Reader (BMG Labtech, Ortenberg, Germany). Each assay plate included controls—DMSO control (0.1%), positive control, negative control, and blanks—with a minimum of 12 wells per plate analyzed. The results are reported as arbitrary luminometric units (ALU).

### 2.3. RNAseq Experiments of Primary Human Hepatocytes and Cell Lines

Total RNA samples were extracted from primary liver cells and cell lines for the RNA sequencing experiments. Trizol was added to each sample, which was then incubated for 5 min and stored at −80 °C until further isolation. The isolation of total RNA was performed using TRIzol reagent (Thermo Fisher Scientific) following the manufacturer’s instructions, including steps with chloroform, isopropanol, and 75% ethanol. The resulting RNA pellet was resuspended in DEPC-treated water. RNA quantification was conducted using a Qubit fluorometer (Thermo Fisher Scientific), with RNA integrity number (RIN) values determined using a TapeStation 2200 (Agilent Technologies, Santa Clara, CA, USA), and an assessment of A260/280 and A260/230 ratios using a NanoDrop^®^ Spectrophotometer ND-1000 (Thermo Fisher Scientific). For library preparation, 0.5–1.0 μg of total RNA with an RIN value greater than 7.0, A260/280 ratio greater than 1.8, and A260/230 ratio greater than 2.0 was utilized. Library preparation was performed using the TruSeq Stranded mRNA kit (Illumina, San Diego, CA, USA). Library quantification was conducted via qPCR and measurements were performed using TapeStation (Agilent Technologies). The RNAseq reads from each sample were aligned to the human reference genome (GRCh38.p14), allowing for a maximum of one mismatch per read. The mapped reads were utilized to assemble known transcripts from the reference, and their abundances were estimated. The expression level of each gene was normalized as fragments per kilobase of exon per million mapped reads (FPKM). To identify significantly differentially expressed genes, the Wilcoxon signed-rank test was employed to compare FPKM expression levels between the RNAseq samples.

### 2.4. scRNAseq of Primary Human Hepatic Cell Populations

Single-cell RNA sequencing (scRNAseq) was conducted on primary human liver cell homogenate populations (OZT) obtained from the healthy donor, UBV. A mixture of hepatic cell populations was carefully washed in Dulbecco’s phosphate-buffered saline (DPBS, 1×; Corning, Cat No. 21-031-CV) and resuspended at a concentration of 10^6^ cells/mL to prevent cell aggregates. The scRNAseq protocol utilized the 10x Genomics Chromium controller, Chromium single-cell 3′ library, and gel bead kit (10x Genomics, Pleasanton, CA, USA, Cat No. PN-1000075) following the standard manufacturer’s protocols. Briefly, gel beads in emulsion (GEMs) were generated by combining barcoded single-cell 3′ v3 gel beads, a master mix containing cells, and partitioning oil onto Chromium chip B. Cells were delivered at a limiting dilution to achieve single-cell resolution, with the majority (~90–99%) of generated GEMs containing no cells. Between 2000 and 21,000 live cells were loaded onto the Chromium controller to recover between 1500 and 15,000 cells for library preparation and sequencing. Following GEM generation, the gel beads were dissolved, primers were released, and any co-partitioned cells were lysed.

An Illumina TruSeq Read 1, 16 nt 10x barcode, 12 nt unique molecular identifier (UMI), and 30 nt poly(dT) sequence was mixed with the cell lysate and a master mix containing reverse-transcription (RT) reagents. The incubation of the GEMs produced barcoded, full-length cDNAs from poly-adenylated mRNAs. Subsequently, the GEMs were broken, and cDNA was amplified and quantified using an Agilent high-sensitivity DNA screentape. SPRIselect magnetic beads were used to purify the first-strand cDNA, followed by PCR amplification to generate sufficient mass for library construction. Enzymatic fragmentation and size selection optimized the cDNA amplicon size, with TruSeq Read 1 added during GEM incubation. P5, P7, a sample index, and TruSeq Read 2 were added via end repair, A-tailing, adaptor ligation, and PCR. Library quality was assessed using an Agilent Bioanalyzer high-sensitivity chip. Sequencing was performed on the Illumina HiSeq 2500 with a target of 150,000 reads/cell. Data processing, sample demultiplexing, and gene expression quantification were performed using the Cell Ranger Single-Cell software Suite v7.2 (10x Genomics). Genes with more than one unique molecular identifier (UMI) count were considered for analysis. The top 1000 most variably expressed genes were used for clustering, and t-distributed stochastic neighbor embedding (t-SNE) analysis was employed to reduce data dimensionality [36]. K-means clustering was used to identify cell populations, with cell classification based on Spearman’s correlation of mean expression profiles.

### 2.5. HCC Clinical Progression Patterns via Transcriptome Data Analysis

The progression of hepatocellular carcinoma (HCC) was analyzed across various stages, ranging from preneoplastic lesions (cirrhosis and dysplasia) to advanced metastatic tumors. The raw data and clinical grading were retrieved from Wurmbach et al. [37]. Samples representing normal tissue, non-tumor cirrhotic lesions, dysplastic nodules of low and high grades, and different stages of HCC (very early, early, advanced, and very advanced) were included in the analysis. A comparison was made between non-tumor stages and the advanced to very advanced HCC stages using the Wilcoxon test, with a Benjamini–Hochberg adjustment of *p*-values. Patterns related to heme biosynthesis, cytochrome production, electron transport chain supply, and heme trafficking were analyzed and normalized on a scale from 1 to 100 for visualization purposes.

### 2.6. Patient Tumor Gene Expression Analysis and Survival Analysis

Gene expression data from liver cancer patients and normal tissues were retrieved from The Cancer Genome Atlas (TCGA), and GETx [38,39] was analyzed to investigate the association between heme biosynthesis and patient survival outcomes. A set of 182 patient samples with higher gene expression levels was compared to a set of 182 patient samples with lower expression levels. The expression levels of key genes related to heme biosynthesis (ALAS1, HMBS, FLVCR1, UQCRH, and CYP2C8) were examined. The MKI67 gene was used a marker of oncogenesis. Kaplan–Meier survival curves [40] and log-rank tests [41] were utilized to assess the correlation between gene expression levels and overall survival (OS). Patients were stratified into high and low gene expression groups, and statistical significance was determined using log-rank tests.

## 3. Results

### 3.1. RNAseq Analysis Reveals Mass Alterations in Heme Metabolism Between Normal Primary Human Hepatocytes and Hepatoma Cell Lines

Human primary hepatic metabolism is highly sensitive to laboratory manipulations and culture adaptations [42]. To investigate heme-metabolism-related gene expression in primary human hepatocytes, RNAseq was performed directly on hepatocytes obtained from two healthy human donors (UVB and PDC) without any culture steps. Our experiments provided an extensive coverage of primary hepatocyte transcriptomes, revealing the expression of approximately 7000 genes in human hepatocytes. Robust gene expression profiles associated with heme biosynthesis pathways were evident in healthy liver samples, facilitating crucial cellular processes such as electron transport chain function, exemplified by UQCRH and cytochrome P450 enzymes, such as CYP2E1 (Figure 1) (Appendix A).

To compare primary human hepatocyte metabolism with hepatoma cell line metabolism, we sequenced two liver cancer cell lines (HC04 and HepG2). RNAseq analysis contrasting primary human hepatocytes from healthy liver donors (UVB and PDC) with hepatoma cell lines (HC04 and HepG2) revealed significant alterations in heme metabolism in oncogenic transformation, as depicted in Figure 1. Among the 6989 genes examined, 26% exhibited downregulation in cancer cell lines indicative of hepatic functional loss. Prominently, genes associated with primary liver metabolism, such as ARG1, UGT2R4, and HMGCS2, showed a downregulation rate of 81% in cancer lines. Conversely, 11% of genes showed upregulation in cancer cell lines associated with oncogenic transformation, such as CDK6 (Log2 fold change of expression >1, adjusted *p* value < 0.05), as indicated by the differentiation markers upregulated by 71% in cancer lines.

Next, the expression of heme-metabolism-related genes was examined. Volcano plots illustrated distinct patterns of differential gene expression, revealing a substantial loss of major P450 cytochromes in liver cancer cell lines, with 79% being downregulated in cancer lines. Components of the electron transport chain exhibited a 50% downregulation in cell lines, albeit to a lesser extent compared to P450, with most exhibiting a log2FC ≤ −2. Among heme biosynthesis enzymes, only the first two-step genes ALAS1 and ALAD were downregulated in liver cancer lines. With ALAS1 being the rate-limiting step of heme biosynthesis [43], our results suggest rewired hepatic heme production. Additionally, heme degradation enzymes displayed an interesting contrasting pattern of isozyme BLVRA and BLVRB gene expression, indicating that normal hepatocytes and liver cancer lines utilize different processes for heme degradation.

Importantly, while in vitro conditions typically exhibited a mild downregulation of the electron transport chain in cell lines, as observed in Figure 1, our further in vivo analyses (described in the following sections), including the single-cell RNAseq (scRNAseq) of patient samples, liver cancer progression biopsy, and tumor tissue expression data, demonstrated sustained or even enhanced electron transport chain expression levels.

In summary, our findings confirm previous studies that liver heme-related drug metabolism is most lost in standard laboratory-cultured cell lines. Additionally, we report transcriptome-related aberrant heme biosynthesis and heme degradation patterns.

### 3.2. Single-Cell Transcriptome Analysis Reveals Heme Rewiring Associated with Cell-to-Cell Heterogeneity in Oncogenesis in Human Liver Cancer

To further investigate the observed alterations in heme metabolism, particularly the rewired heme biosynthesis and significant downregulation of cytochrome P450 expression identified in our bulk RNAseq experiments, we analyzed a set of previously published single-cell RNA sequencing (scRNAseq) data [44,45] from human hepatocellular carcinoma (HCC) patients. These analyses were aimed at studying the details of the cell-to-cell heterogeneity related to the rewiring of heme metabolism associated with cancer aggressiveness within tumors.

The patient-derived liver cancer single cells were initially reclustered into two distinct groups, more differentiated vs. more transformed, as illustrated in Figure 2A. The segregation was based on the hierarchical clustering of differentially expressed genes, with HCC_D representing more differentiated cells and HCC_T representing a more transformed (aggressive) cell population. The subsequent examination of heme biosynthesis and metabolic processes in the scRNAseq expression profiles of the HCC samples revealed several key findings (Figure 2B) (Appendix A). Firstly, genes specific to in vivo oncogenesis in the HCC_T group (absent in the HepG2 cell lines) highlighted S100A family genes as early oncogenesis markers, consistent with previous conclusions [46]. Furthermore, both HCC_T and HepG2 exhibited an overall loss of expression in the cytochrome P450 gene family, indicating a commonality in this alteration across different liver cancer models. However, genes encoding electron transport chain (ETC) hemoproteins remained largely unchanged during in vivo oncogenesis, suggesting that the mild downregulation of ETC in cancer lines may be an artifact of laboratory conditions.

Similar to our bulk RNAseq experiment results [31,47], the single-cell RNAseq analysis (Figure 2C) revealed the downregulation of the first- and second-step heme biosynthesis genes ALAS1 and ALAD, respectively, in the more aggressive HCC_T cell population. This observation was consistent with the downregulation pattern observed in our primary human hepatocyte RNAseq experiments (Figure 2D), indicating rewired heme biosynthesis in both liver cancer lines in vitro and in more transformed cancer cells in patient samples.

### 3.3. Heterogeneous Distribution of Heme Metabolism across Normal Human Liver Cell Populations in scRNAseq

Using scRNAseq, we studied the diverse landscape of heme metabolism within different cell populations of normal human liver tissue from donor UBV. Given the crucial role of liver macrophages in heme homeostasis [48], we enriched for the macrophage Kupffer cell marker CD163 to ensure the accurate identification of liver-resident macrophages. In total, we generated transcriptomes of 1170 high-quality single cells using methods similar to our previous study [28]. Subsequent manifold learning and clustering analysis revealed six distinct cell populations (Figure 3A) (Appendix A), with significant differentially expressed genes identified for further cell-type characterization and functional annotations [49,50] (Appendix A). For each cluster of significantly differentially expressed genes defining the single-cell populations, we conducted cell-type identification studies using predefined marker sets and non-parametric testing (Figure 3B). Distinct populations in the liver of UBV were identified, including hepatocytes (enriched in *APOC2* gene expression), Kupffer cells (enriched in *CD163* gene expression), T cells (enriched in *IL7R* and *CD8A* gene expression), and NK cells (enriched in *CD160* and *GZMB* gene expression).

Our gene expression analysis focused on heme-metabolism-related genes across the identified cell populations (Figure 3C). The rate-limiting step gene *ALAS1*, indicative of heme biosynthesis activity, exhibited significant upregulation in hepatocytes. Additionally, significant differences in cytochrome production were observed in hepatocytes, suggesting their dominant role in heme biosynthesis and cytochrome P450 supply. In contrast, Kupffer cells displayed an upregulated gene expression in the process of heme breakdown, implying their role in heme recycling and homeostasis regulation. Furthermore, no significant differential expression was observed in genes related to the ETC across these populations, highlighting the importance of ETC function across all liver cell types.

To investigate heme homeostasis in hepatic cell proliferation within healthy livers, we conducted further analysis by categorizing single cells into potentially proliferating and non-proliferating populations based on established proliferation markers [49,50] (Figure 3D) (Appendix A). Our findings indicated no significant differences in heme biosynthesis activity or hemoprotein gene expression between proliferating and non-proliferating cells within healthy human liver tissue. This suggests the robust and efficient maintenance of heme homeostasis across various liver cell functions.

### 3.4. Correlation Between Heme Metabolic Shift and Clinical Progression in Liver Cancer Patient Gene Expressions

To study the clinical relevance of heme metabolism in cancer progression, we analyzed previously generated high-quality liver cancer progression gene expression data from 75 patient liver tissues [37]. These were indicative of dynamic metabolic rewiring during disease advancement. We examined the progression of hepatocellular carcinoma (HCC) across various stages, from preneoplastic lesions (cirrhosis and dysplasia) to advanced end-stage tumors, as defined by Wurmbach et al. [37]. Our gene expression pattern analysis included samples representing normal tissue (i.e., normal), non-tumor cirrhotic lesions (i.e., cirrhosis_1), dysplastic nodules of low and high grades (i.e., cirrhosis 2 and 3), and different stages of HCC (i.e., HCC1–4), enabling a comparison to assess the statistical significance of observed gene expression changes in the sequence of oncogenesis. While the current data do not enable a refined classification of HCC subtypes, the significant patterns observed in tumor staging provide valuable opportunities for further investigation. Figure 4A schematically illustrates the major differences in heme metabolism between healthy and oncogenic livers undergoing HCC progression. While robust heme biosynthesis in healthy liver tissues supports vital metabolic functions (including P450 metabolism, essential for cellular homeostasis), liver cirrhosis and liver cancer exhibit a marked reduction in heme biosynthesis and hemoprotein production, coupled with sustained or enhanced ETC activity.

A detailed analysis of patient liver cancer progression transcriptome data revealed distinct changes in gene expression profiles across disease stages (Figure 4B) (Appendix A). As expected, markers of liver differentiation, such as *EPHX2* [51], showed downregulation, while those associated with cancer proliferation, such as *MKI67*, increased. Notably, *ALAS1*, the rate-limiting first-step gene in heme biosynthesis, was downregulated, while the mid-pathway heme biosynthesis gene *HMBS*, driving intermediate porphyrin accumulation [28], showed opposing increasing expression during hepatic function degeneration, supporting the porphyrin overdrive hypothesis in oncogenesis. The expression of the final-step enzyme-encoding gene FECH was observed to be repressed during the early stages of liver pathology. However, the overall expression trend did not reach statistical significance (*p* > 0.05). Despite the crucial function of FECH in heme biosynthesis and the expected dysregulation in liver pathology, our current analysis does not allow for definitive conclusions. This limitation may be attributed to constraints within our dataset.

Our observations of skewed expression in key heme biosynthesis genes, specifically the genes encoding the first-step enzyme ALAS1 and the mid-pathway enzyme HMBS, suggest an imbalanced pathway that may lead to the accumulation of intermediates. This interpretation aligns with findings from our recent studies [28,52] where we directly observed protoporphyrin IX (PPIX) accumulation in solid tumor cell lines. Additionally, these observations are consistent with clinical practices involving photodynamic therapy (PDT) for treating liver cancers [53,54,55,56], which utilize the light activation of tumor-accumulated porphyrins to destroy hepatoma cells.

Porphyrins are oxidized macrocycles, and PPIX is the primary porphyrin (penultimate heme intermediate) in the heme biosynthetic pathway. Other types, such as coproporphyrin isomers, may appear as off-pathway downstream metabolic products derived from early intermediates like coproporphyrinogen. The existing literature predominantly identifies PPIX as the major accumulated form in both in vitro and in vivo cancer cells [57,58,59,60]. In addition our gene expression and forward genetic data analysis, future studies will be required to verify the exact biochemical identity of heme precursors involved in oncogenesis in liver tissues.

Key P450 genes, such as *CYP2A6* and *CYP2C8* [17], displayed reduced expression levels in cancer progression, consistent with our bulk RNAseq and scRNA-seq results, while ETC-related genes, such as *CYC1* and *COX7B2*, were upregulated during cancer progression, confirming that in vivo human liver tumorigenesis sustains or boosts the ETC. These heme metabolic gene expression pattern shifts indicate adaptive responses occurring within cancerous tissues, potentially contributing to disease progression by prioritizing heme supply to different processes demanding heme as a cofactor.

The clinical relevance of these metabolic alterations is evident in their correlation with patient overall survival (OS). Kaplan–Meier analysis derived from the TCGA project demonstrated that heme biosynthesis and P450 expression were associated with inferior survival outcomes, while enhanced protoporphyrin IX (PPIX) accumulation and trafficking, along with increased ETC production, were linked to worse survival outcomes. Notably, the increased expression of the heme export gene FLVCR1 [34,61] was found in tumor progression (Figure 4B), and associated with poor survival (Figure 4C), indicating a significant role of heme and porphyrin trafficking in shaping the tumor microenvironment.

Together, these clinical transcriptome analyses, encompassing both HCC progression gene expression and patient survival data, show the potential of heme metabolic markers as prognostic indicators in liver cancer, suggesting the role of heme metabolism in driving disease progression.

### 3.5. Targeting Liver Cancer Porphyrin Overdrive In Vitro with a Bait-and-Kill Strategy

Our genomic analyses using bulk, single-cell, and clinical transcriptomic data highlight the need for future detailed biochemical investigations to identify the specific metabolites involved. Nonetheless, our study shows the potential of heme metabolic markers as prognostic indicators in liver cancer, offering insights into therapeutic strategies. Here, we demonstrate the feasibility of targeting porphyrin overdrive in liver cancers through an in vitro proof-of-principle drug–drug synergy study. Building upon our lab’s findings and extensive previous research showing that cancer cell lines accumulate protoporphyrin IX (PPIX) [28,57,58,59,60], we have developed a bait-and-kill strategy [28] targeting cancer porphyrin overdrive. This strategy utilizes the inducibility of PPIX by 5-aminolevulinic acid (ALA) and exploits the redox reactivity [62] of PPIX. The approach involves ‘baiting’ cancer cells with ALA to induce PPIX accumulation, followed by ‘killing’ the cells with a compound that leverages the metabolic stress from the accumulated porphyrins. This method harnesses the resulting accumulation of redox-active porphyrins [63,64] to effectively induce cell death. Artemisinin, an antimalarial drug known to be activated by heme metabolites [65], is hypothesized to synergize with the induced PPIX accumulation to effectively eradicate cancer cells.

We conducted experiments to quantify the cytotoxic effects of ALA and dihydroartemisinin (DHA), both individually and in combination, across a broad range of drug concentrations, as shown in Figure 5, using two different human liver cancer cell lines. Drug–drug interaction studies were performed in triplicate experimental setups. Our findings indicate that while ALA and DHA alone exhibited minimal toxicity, their combined administration demonstrated significant synergy in killing cancer cells. Remarkably, this synergistic effect was observed even at low micromolar concentrations of DHA, suggesting that ALA enhances DHA’s efficacy through PPIX accumulation, thereby achieving effective cancer cell clearance in vitro.

## 4. Discussion

In this study, we have inferred the metabolic partitioning of heme metabolism within the liver, a crucial organ in human metabolism. Specifically, we mapped out the gene expression pathways through which heme is utilized in normal liver function, demonstrating its roles in both the electron transport chain (ETC) and cytochrome P450 (P450) enzymes. Notably, our transcriptome analysis revealed that normal liver heme homeostasis is highly efficient, as evidenced by the absence of intermediate accumulations in any cell population, including proliferating cells. Furthermore, we highlighted the specialized role of liver Kupffer cells in the breakdown of heme within normal liver tissue, as also observed in other studies [48,66,67]. This metabolic mapping provides insights into the processes governing heme utilization and degradation within the liver.

Our transcriptome study infers a significant metabolic priority shift in heme metabolism during liver oncogenesis, marked by a pronounced and early suppression of P450 activity. This abrupt reduction in P450 expression reflects a rapid loss of metabolic function within the liver during cancer progression [37,44]. In contrast, we observed a sustained and even enhanced supply to the electron transport chain (ETC) throughout the clinical stages of patient progression. This finding suggests that despite the metabolic dysfunction associated with liver cancer, tumors exhibit robust energy production mechanisms and a secured heme supply, likely promoting their survival and proliferation. However, we caution against interpreting ETC enhancement as direct evidence of a canonical mitochondrial-based oxidative phosphorylation (OXPHOS) pathway; it could also signify an aberrant form of cancer bioenergy production [68] (unpublished data). As the field of tumor bioenergetics rapidly evolves [69,70,71], a clearer picture will likely emerge in the future.

The accumulation of porphyrin intermediates in liver cancer presents an intriguing yet poorly understood phenomenon [28,72]. In this study, transcriptome data show that under normal physiological conditions, the liver efficiently produces large quantities of heme without the accumulation of intermediates, as phenotypically shown in our previous publication [28]. It is perplexing why tumors specifically accumulate these intermediates and why the number escalates with cancer progression, especially considering that no known physiological role is attributed to these intermediates. Additionally, the observation of enhanced heme/porphyrin trafficking suggests a potential role in orchestrating the metabolic microenvironment of the tumor. Our drug synergy data show that it is possible to exploit porphyrin overdrive in liver cancers as a potential target. Further research is warranted to elucidate the precise mechanisms underlying porphyrin accumulation in liver cancer and its implications for tumor metabolism and progression.

## 5. Conclusions

In conclusion, our study shows the importance of heme and porphyrin metabolism in both maintaining liver health and cancer progression. Through analyses of primary human liver transcriptomics and single-cell RNA sequencing, we have inferred significant disparities in heme biosynthesis and analysed its downstream effects. Our findings highlight the importance of precise heme homeostasis in maintaining normal liver function and infer the disruptive consequences of dysregulated heme metabolism in cancer, particularly characterized by the loss of P450 expression, enhanced electron transport chain activity, and porphyrin accumulation.

A limitation of our study is its reliance on bulk, single-cell, and clinical transcriptomic data, necessitating future detailed biochemical investigations to elucidate specific metabolite species involved. Nevertheless, our findings provide insights for potential therapeutic strategies. We demonstrate the feasibility of targeting porphyrin overdrive in liver cancers through an in vitro proof-of-principle drug–drug synergy study. 

## Figures and Tables

**Figure 1 biomolecules-14-00959-f001:**
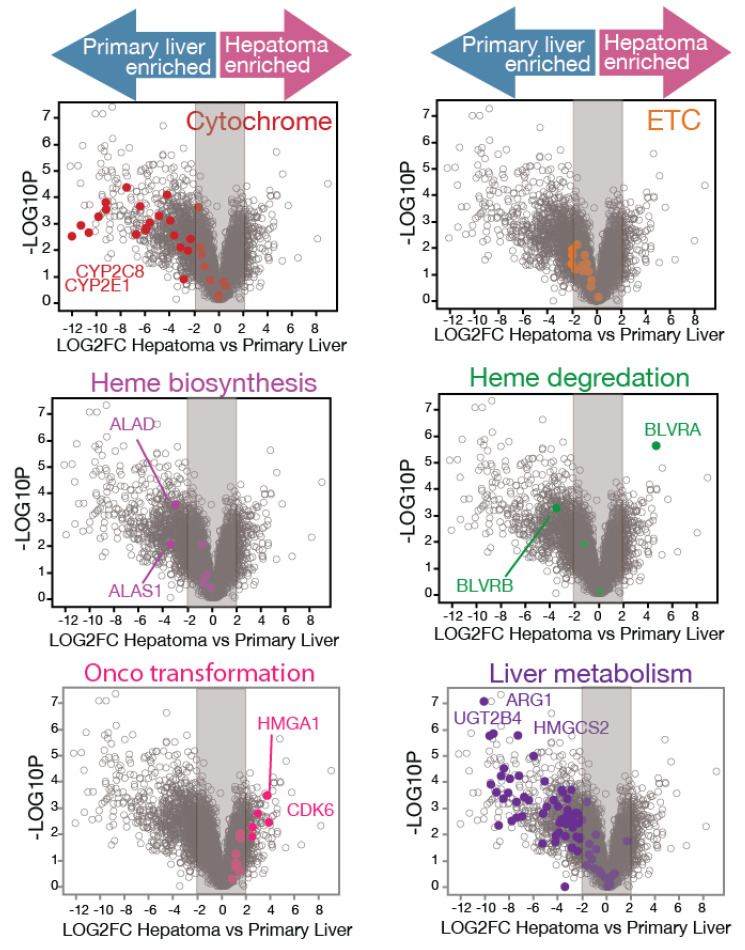
Differential heme metabolism in normal human hepatocytes vs. cancerous liver cell lines, analyzed by RNAseq. RNAseq experiments compared primary human hepatocytes from healthy liver donors (UVB and PDC) with hepatoma cell lines (HC04 and HepG2), revealing significant differences in heme metabolism and oncogenic transformation. Analysis of 6989 genes demonstrated that 26% were downregulated and 11% were upregulated in cancer cell lines (Log2 fold change of expression >1, adjusted *p* value < 0.05). Volcano plots depict differential gene expression with six distinct gene groups color-coded. Cytochromes primarily include genes encoding cytochrome P450 enzymes, crucial for drug metabolism, hormone synthesis, and detoxification, with a 79% downregulation in cancer lines indicating a reduced capacity for these processes. Components of the electron transport chain (ETC) involve genes essential for ATP production via oxidative phosphorylation, with a 50% downregulation suggesting compromised energy metabolism in cancer cells. Heme biosynthesis enzymes focus on the initial steps of heme production, specifically ALAS1 and ALAD, whose downregulation can limit heme-dependent processes. Heme degradation enzymes feature contrasting expressions of isozymes like heme oxygenases, indicating shifts in heme homeostasis and oxidative stress management. De-differentiation markers, upregulated by 71% in cancer lines, are associated with a loss of cell differentiation and an increase in stem-cell-like properties. They are indicative of tumor properties. Lastly, primary liver metabolism genes, crucial for core metabolic function, were downregulated by 81%, highlighting significant disruption in the metabolic capabilities of liver cancer cells. Representative gene symbols are labeled in each group. The Y-axis represents the significance of differential expression as −log10P, calculated using the Wilcoxon test with Benjamin–Hochberg adjustment. Genes enriched in primary liver cells (downregulated in hepatoma cell lines) are depicted on the left side, while genes enriched in hepatoma cell lines (upregulated in cancer cell lines) are depicted on the right side. Note that the mild downregulation of ETC in cell lines is restricted to in vitro conditions; in vivo, maintained or enhanced ETC expression is found in the scRNAseq of patient samples, patient liver cancer progression biopsy, and patient tumor tissue expression data.

**Figure 2 biomolecules-14-00959-f002:**
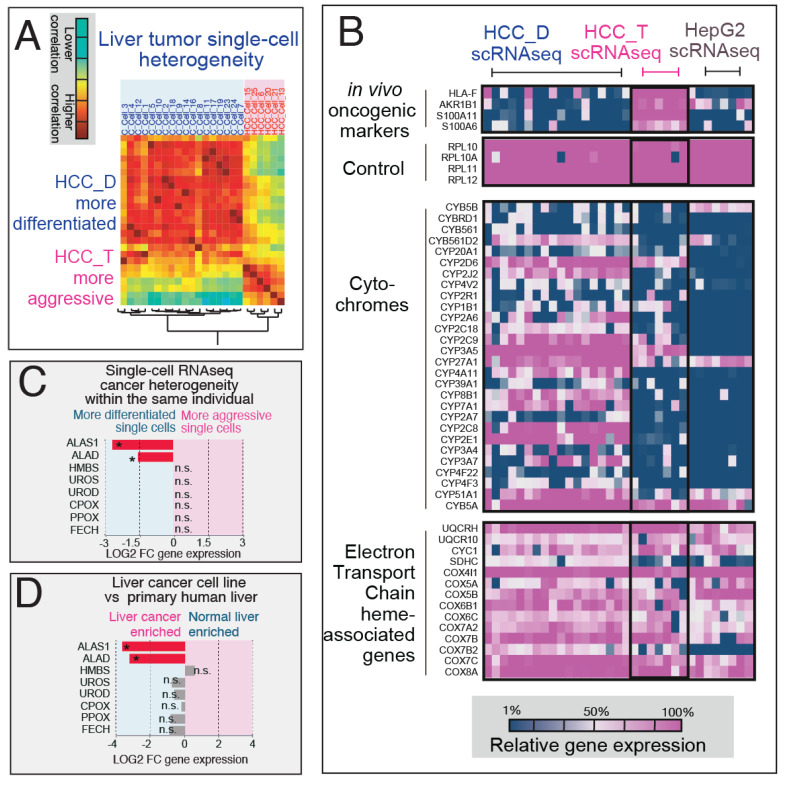
scRNAseq analysis of heme and porphyrin metabolism in liver cancer patient samples. The scRNAseq data from human hepatocellular carcinoma (HCC) patients were reanalyzed based on genes encoding heme metabolism and cell transformation markers. The raw data were sourced from [45]. (**A**) HCC single cells were visualized into two distinct groups: HCC_D representing more differentiated single cells, and HCC_T representing a more transformed (aggressive) single-cell population, as observed through clustering distance matrix based on Pearson’s pair-wise correlation-derived distance similarities of each single-cell expression profile. (**B**) Heatmap visualization of scRNAseq expression profiles of HCC samples. HCC_T expressed a set of genes specific to in vivo oncogenesis that were absent in HepG2 cell lines, with ribosomal genes used as controls. Both HCC_T and HepG2 show an overall loss of expression in the cytochrome P450 gene family. However, genes encoding electron transport chain hemoproteins remain largely unchanged during both in vivo and in vitro oncogenesis. (**C**) Within the same individual, single-cell RNAseq reveals downregulation of the first- and second-step heme biosynthesis genes ALAS1 and ALAD, respectively, in the more aggressive cell population (HCC_T). The subsequent step genes (step 3–step 8) of the heme biosynthesis pathway were not found to be significantly differentially expressed. * indicates significant differential expression determined by Wilcoxon test with BH adjustment. (**D**) A similar pattern of downregulation in the first- and second-step genes was observed in our primary human hepatocyte RNAseq experiments. n.s. indicates not significant.

**Figure 3 biomolecules-14-00959-f003:**
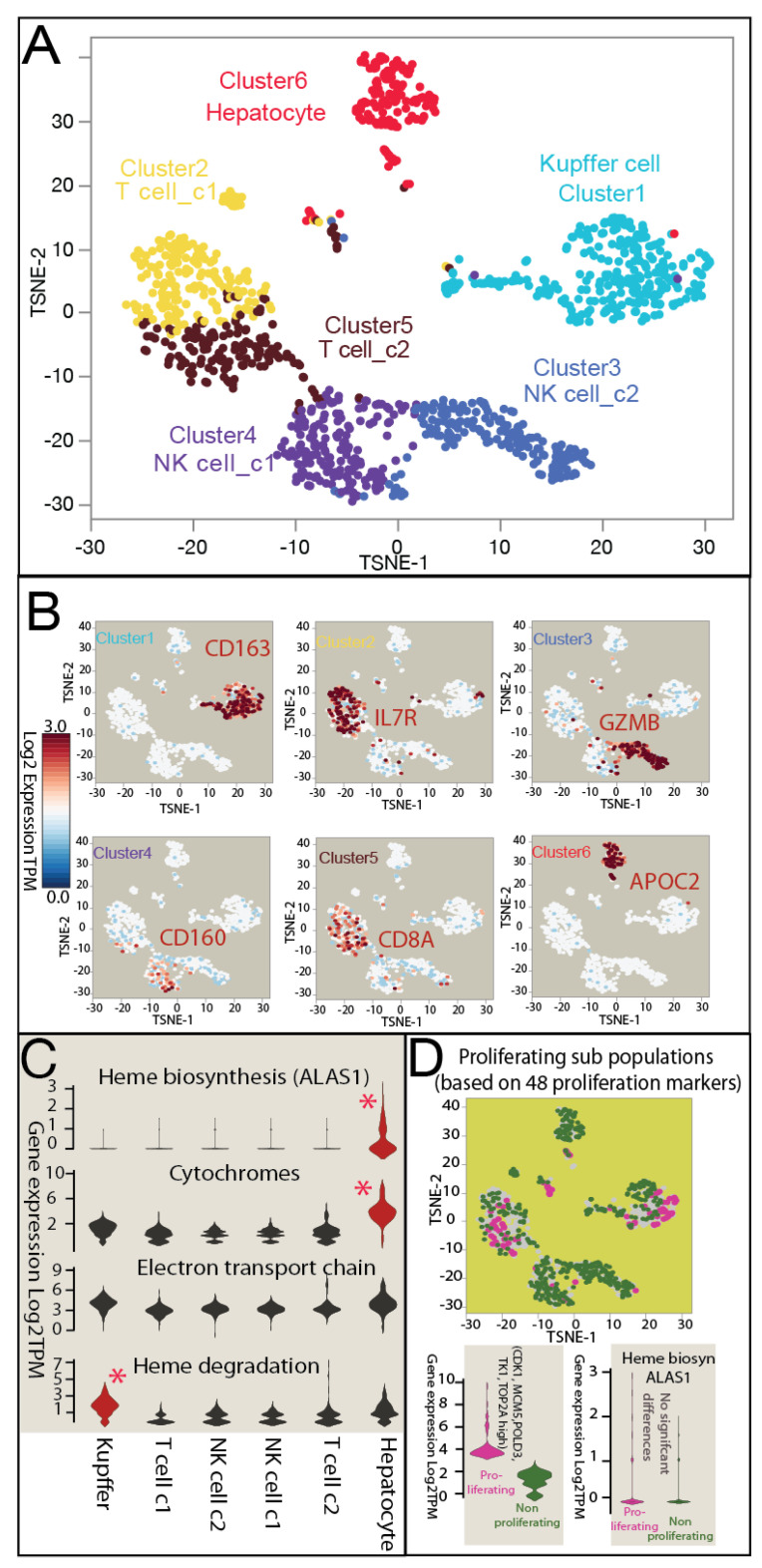
scRNAseq of normal human liver cell populations of hepatocytes and immune cells. (**A**) A total of 1170 single cells with high-quality transcriptomes were utilized for manifold learning and clustering analysis. Six distinct cell populations were identified through t-SNE (t-distributed stochastic neighbor embedding) analysis, including hepatocytes, Kupffer cells (liver-resident macrophages), T cells, and NK cells. Macrophage Kupffer cell marker CD163 was enriched to ensure the capture of liver-resident macrophages. (**B**) Marker genes of each cluster are plotted on the scRNAseq map to visualize the range and extent of gene expression in different cell populations. (**C**) Genes encoding proteins for heme metabolism were analyzed in the cell populations. The rate-limiting step gene of heme biosynthesis, ALAS1, was used to show heme biosynthesis. The full set of cytochrome genes depicted P450 production, while the set of genes encoding electron transport chain (ETC) hemoproteins represented ETC-related heme production. Additionally, a set of four genes encoding heme oxygenases and bilirubin degradation enzymes indicated heme degradation processes. * and red shading denote significant differences in heme biosynthesis and cytochrome production in hepatocytes, and a significantly upregulated process of heme breakdown in Kupffer cells. ETC-related genes were not significantly differentially expressed. (**D**) The full set of single cells was divided into potentially proliferating and non-proliferating cell populations based on a set of proliferation markers, marked by pink (potentially proliferating) and green (potentially non-proliferating) cells. Heme biosynthesis was not found to be significantly different for proliferating cells in healthy human liver.

**Figure 4 biomolecules-14-00959-f004:**
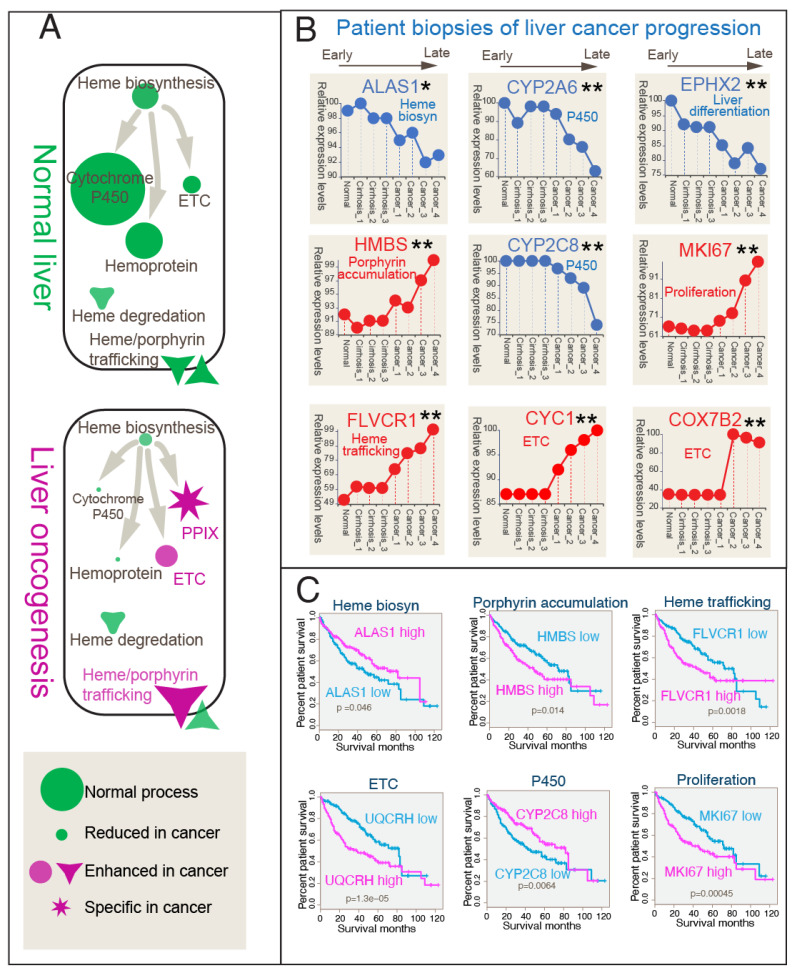
Liver oncogenesis progression gene expression analysis and patient overall survival outcomes related to heme metabolism. (**A**) Schematic illustration of heme metabolism in normal vs. oncogenic livers. In normal livers, robust heme biosynthesis supports a large quantity of P450 metabolism and other hemoproteins. In oncogenic livers, heme biosynthesis and hemoprotein production are reduced, but the electron transport chain (ETC) remains similar or enhanced in liver cancers. Protoporphyrin IX (PPIX) production is specific to the liver. Heme/PPIX trafficking is enhanced in liver cancers. (**B**) Patient biopsies in liver cancer progression transcriptome data show a heme metabolic priority shift during hepatocellular carcinoma (HCC) progression. Different stages of disease development from mild cirrhosis to severe cirrhosis (cirrhosis 1–3) and early- to late-stage HCC are plotted (cancer 1–4). The data were reanalyzed from [37]. The liver differentiation marker EPHX2 shows downregulation during progression, whereas the cancer proliferation marker MKI67 shows an increase. The major P450 genes CYP2A6 and CYP2C8 show a loss of expression with cancer progress, whereas the ETC-related genes CYC1 and COX7B2 show an increase with liver cancer development. The mid-pathway heme biosynthesis gene HMBS, linked to PPIX production, is upregulated. Heme/PPIX export is upregulated. * denotes significant changes between early and late stages (*p* < 0.05). (**C**) Liver cancer heme overdrive is linked to patient overall survival (OS). Heme overdrive relates to OS. The Kaplan–Meier estimator was used on the lifetime data (from TCGA liver cancer). Log rank tests were conducted. Heme biosynthesis and P450 are linked to inferior survival, while enhanced PPIX accumulation and trafficking, in addition to ETC production, are linked to worse survival outcomes. ** denotes higher level of significance with (*p* < 0.001).

**Figure 5 biomolecules-14-00959-f005:**
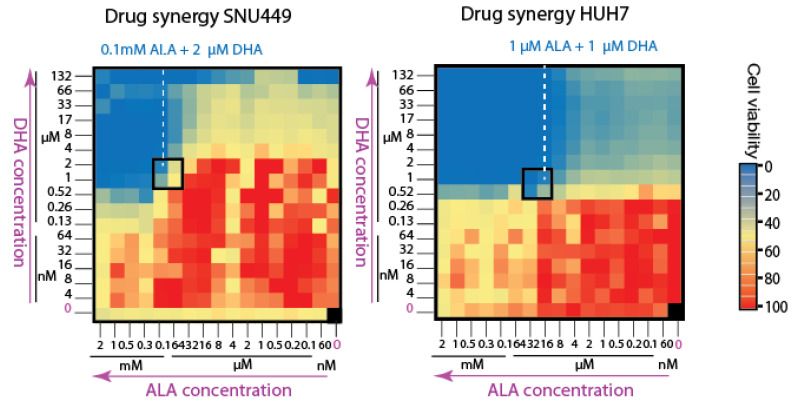
Drug synergy to target porphyrin overdrive in liver cancer cell lines. The bait-and-kill strategy utilized a two-step drug synergy approach: first, 5-aminolevulinic acid (ALA) was used as a bait to induce protoporphyrin IX (PPIX) accumulation, and second, dihydroartemisinin (DHA) was employed as a killing agent to synergize with ALA to eliminate cancer cells. Drug synergy was evaluated in two human liver cancer cell lines: SNU449 and HUH7. High doses of ALA alone failed to clear cancer cells in either cell line. However, when used in combination, DHA in the micromolar (μM) range effectively cleared all cancer cells with less than 0.2 mM ALA. Cell viability was assessed using CellTiter-Glo^®^ 2.0 reagent to measure cellular ATP levels, with 25.6 μL of reagent added per well. Luminescence was quantified using a Clariostar Plus Microplate Reader.

## Data Availability

The datasets generated and analyzed during the current study are available in the NCBI/GEO repository GSE268462.

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
