# Peer review of "Harnessing Porphyrin Accumulation in Liver Cancer: Combining Genomic Data and Drug Targeting"

_biomolecules, 2024, doi:10.3390/biom14080959_

Round 1

Reviewer 1 Report

Comments and Suggestions for Authors

This manuscript describes that disruption of heme biosynthesis promotes the accumulation of heme intermediates, which may enhance tumor survival. The authors used primary human liver transcriptomics and single-cell RNA sequencing to investigate heme and porphyrin metabolism in healthy and carcinogenic human livers. The study revealed robust gene expression patterns in healthy liver for heme biosynthesis, supporting ETC and CYP450 function, without accumulation of intermediates. Gene expression showed that as patients’ cancer progressed, the heme supply to the ETC remained largely unchanged or even increased, suggesting a shift in metabolic priorities. In addition, the authors' genomic findings confirmed a link between abnormal expression of genes involved in porphyrin metabolism and poor overall survival in aggressive cancers, providing potential targets for clinical treatment development. All in all, the author did an excellent job. The manuscript is also well writing and represents an interesting work. As such, I recommend this manuscript to be published after minor revision as below:

1.     Figure 1, please provide more explanationts;

2.     References, journal abbreviations or full names, need to be consistent;

3.     References, articles should use the same capitalization for the first letter of each word;

4.     References need to be checked uniformly;

Author Response

Reviewer #1

This manuscript describes that disruption of heme biosynthesis promotes the accumulation of heme intermediates, which may enhance tumor survival. The authors used primary human liver transcriptomics and single-cell RNA sequencing to investigate heme and porphyrin metabolism in healthy and carcinogenic human livers. The study revealed robust gene expression patterns in healthy liver for heme biosynthesis, supporting ETC and CYP450 function, without accumulation of intermediates. Gene expression showed that as patients’ cancer progressed, the heme supply to the ETC remained largely unchanged or even increased, suggesting a shift in metabolic priorities. In addition, the authors' genomic findings confirmed a link between abnormal expression of genes involved in porphyrin metabolism and poor overall survival in aggressive cancers, providing potential targets for clinical treatment development. All in all, the author did an excellent job. The manuscript is also well writing and represents an interesting work. As such, I recommend this manuscript to be published after minor revision as below:

Response:

We greatly appreciate the reviewer’s positive feedback and the thoughtful summary of our work. We are pleased that the reviewer found our manuscript interesting and valuable to the field. Following the recommendations, we have carefully considered the suggestions and have made the appropriate revisions to the manuscript. We look forward to the opportunity to contribute to the literature with this work.

  1. Figure 1, please provide more explanations;

Response:

We thank the reviewer for the feedback regarding Figure 1. We have enhanced the explanation for "Figure 1: Differential Heme Metabolism in Normal Human Hepatocytes vs. Cancerous Liver Cell Lines by RNAseq." In the revised figure legend, we now include a more detailed description of the function and expression profiles of each gene subgroup compared in the analysis. This addition aims to provide a clearer understanding of the differences in heme metabolism between normal and cancerous liver cells, as demonstrated through RNA sequencing.

  1. References, journal abbreviations or full names, need to be consistent;

Response:

We have used abbreviations for journal names consistently in the revision.

  1. References, articles should use the same capitalization for the first letter of each word;

Response:

We have used sentence case for references in the revision.

  1. References need to be checked uniformly;

Response:

We have check the full set of references and made adjustment for consistency

Reviewer 2 Report

Comments and Suggestions for Authors

This manuscript compares the gene expression of cancer liver cells with that of normal liver cells. The authors found that the expression of heme metabolism-related genes is altered in cancer liver cells, suggesting that heme and/or its intermediates, including protoporphyrin IX (PPIX), play a key role in cancer progression. The text is well organized and the data presented are clear. References are properly cited. I believe this manuscript will be of interest to the readership of Biomolecules The reviewer is willing to accept this manuscript for publication after some minor revisions.

Minor points,

1. Lines 328-330: I understand that the expression of ALSA1 is decreased during cancer progression, while that of HMBS is increased. However, these results do not necessarily lead to the accumulation of the intermediate porphyrin. Is there any other direct observation of its accumulation?

2. Lines 331-333: Although the space is limited, I would like to see the expression level of FECH in Fig. 4B.

Author Response

Reviewer #2

This manuscript compares the gene expression of cancer liver cells with that of normal liver cells. The authors found that the expression of heme metabolism-related genes is altered in cancer liver cells, suggesting that heme and/or its intermediates, including protoporphyrin IX (PPIX), play a key role in cancer progression. The text is well organized and the data presented are clear. References are properly cited. I believe this manuscript will be of interest to the readership of Biomolecules The reviewer is willing to accept this manuscript for publication after some minor revisions.

Response:

We greatly appreciate the positive feedback and constructive suggestions provided by the reviewer. We have carefully considered and implemented the recommended minor revisions to enhance the clarity and depth of our manuscript. We are happy for the opportunity to contribute to Biomolecules and look forward to potential future contributions.

Minor points,

  1. Lines 328-330: I understand that the expression of ALSA1 is decreased during cancer progression, while that of HMBS is increased. However, these results do not necessarily lead to the accumulation of the intermediate porphyrin. Is there any other direct observation of its accumulation?

Response:

We appreciate the reviewer's observation regarding the expression changes in ALAS1 and HMBS and their implications for porphyrin accumulation. Indeed, the decrease in ALAS1 expression coupled with an increase in HMBS suggests potential imbalances in the heme biosynthesis pathway, which could hypothetically lead to the accumulation of intermediate porphyrins. However, these gene expression changes alone do not confirm the actual accumulation of intermediates.

To address this, we have provided direct evidence of porphyrin accumulation both in liver cancer cell lines, as shown in Figure 5, and through previously published work by Adapa et al., Life Science Alliances, 2024. Additionally, the clinical application of photodynamic therapy (PDT) for liver cancer, which directly exploits the selective accumulation of porphyrins in malignant tissues while sparing healthy surrounding tissue, further supports our findings. We have included this information in the revised manuscript to clarify the direct observations supporting porphyrin accumulation during liver cancer progression.

  1. Lines 331-333: Although the space is limited, I would like to see the expression level of FECH in Fig. 4B.

Response:

We appreciate the reviewer's insightful comment regarding the expression level of ferrochelatase (FECH). In our analysis of the dataset, FECH expression did not show a significant pattern of change across different stages of liver cancer progression. Consequently, including its trend in Fig. 4B might potentially mislead and overstate our findings. We recognize the importance of FECH as the terminal enzyme in the heme biosynthesis pathway. Although we noted a decrease in FECH expression at the early stages of liver fibrosis, which aligned with our expectations, this pattern did not achieve statistical significance overall—likely reflecting limitations in our dataset.

We have added further clarification in the revised manuscript to better explain the context and limitations of these findings, ensuring a more accurate interpretation of the data.
